# Total Lipids, Fatty Acid Composition, Total Cholesterol and Lipid-Soluble Antioxidant Vitamins in the *longissimus lumborum* Muscle of Water Buffalo (*Bubalus bubalis*) from Different Production Systems of the Brazilian Eastern Amazon

**DOI:** 10.3390/ani12050595

**Published:** 2022-02-27

**Authors:** Jamile Andréa Rodrigues da Silva, Laurena Silva Rodrigues, José de Brito Lourenço-Júnior, Cristina Mateus Alfaia, Mónica Mendes Costa, Vinícius Costa Gomes de Castro, Andréia Santana Bezerra, André Martinho de Almeida, José António Mestre Prates

**Affiliations:** 1Institute of Animal Health and Production, Federal Rural University of the Amazônia (UFRA), Belem 66077-830, Brazil; vinicius.c.gomes@hotmail.com; 2Postgraduate Program in Animal Science (PPGCAN), Institute of Veterinary Medicine, Federal University of Para (UFPA), UFRA, Brazilian Agricultural Research Corporation (EMBRAPA), Castanhal 68746-360, Brazil; laurenazootec@gmail.com (L.S.R.); joselourencojr@yahoo.com.br (J.d.B.L.-J.); andreia.silva@ufpa.br (A.S.B.); 3Center for Interdisciplinary Research in Animal Health (CIISA), Faculty of Veterinary Medicine, University of Lisboa, 1300-477 Lisboa, Portugal; cpmateus@fmv.ulisboa.pt (C.M.A.); monicacosta@fmv.ulisboa.pt (M.M.C.); japrates@fmv.ulisboa.pt (J.A.M.P.); 4Linking Landscape, Environment, Agriculture and Food (LEAF), Institute of Agronomy (ISA), University of Lisboa, 1349-017 Lisboa, Portugal; aalmeida@isa.ulisboa.pt

**Keywords:** cholesterol, fatty acids, meat, vitamin A, vitamin E, water buffalo

## Abstract

**Simple Summary:**

The constant search for higher living standards and better health makes consumers more aware of the nutritional compositions of foods. Water buffalo meat has so far been poorly characterized. In this work, we have analyzed the influence of different production systems in the Eastern Amazon on meat nutritional quality parameters of water buffalo. Meat nutritional value was improved when buffaloes were reared in Marajó Island extensive system during the rainy season, as it had lower levels of cholesterol and higher amounts of α-tocopherol, associated with a higher h/H ratio and a lower index of atherogenic. Also, this meat had lower levels of saturated fatty acids and higher amounts of mono- and polyunsaturated fatty acids. Overall, meat from buffaloes raised in pasture production systems showed better lipid nutritional quality.

**Abstract:**

The aim of this study was to analyze the influence of distinct production systems and seasonal variation in the Brazilian Eastern Amazon on the meat lipid composition of water buffaloes. Water buffaloes were reared in commercial farms in the Eastern Amazon either in extensive systems (Marajó Island, Nova Timboteua and Santarém locations), during rainy or dry seasons, or intensive (feedlot) systems. Animals reared in extensive systems were fed natural pastures, and those reared in feedlots were fed sorghum silage and commercial pellets. Buffaloes were slaughtered and ribeye muscle (*longissimus lumborum)* samples collected. Lipid-soluble antioxidant vitamins and fatty acids were analyzed. The nutritional value of meat from buffaloes reared in Marajó Island extensive system during the rainy season was higher than that of other systems, as it had lower levels of cholesterol and higher amounts of α-tocopherol associated with higher hypocholesterolaemic/hypercholesterolaemic ratio and lower index of atherogenic. Also, this meat had lower percentages of saturated fatty acids and higher proportions of mono- and polyunsaturated fatty acids (PUFA), particularly n-3 PUFA, with increased PUFA/saturated fatty acids ratio and decreased n-6/n-3 PUFA ratio. However, all extensive systems produced meat with a relatively low index of thrombogenicity values, which is advantageous for human health.

## 1. Introduction

In the Eastern Amazon region in Brazil, water buffaloes are, generally, extensively reared in pasture-based feeding systems, using four distinct pasture production systems: native pastures on the floodable lands of the Marajó Island; native pastures on the flood lands of the Lower and Mid Amazon micro-regions; native pastures on dry lands; and implemented pastures on dry lands in protected forest areas. In the latter production system, more productive pasture plants are used in herds formed by animals of higher genetic value and production traits [1,2,3]. More recently, intensive buffalo rearing systems have been developed in order to promote faster and more homogenous water buffalo meat production. Currently, the Brazilian water buffalo herd consists of 1.5 million heads with over 1 million (73%) being raised in the Amazon region [4].

The constant search by consumers for higher living standards and health make them more aware of nutritional compositions of foods. There is thus a growing concern about the fat and cholesterol content of animal products. It is well known that ruminant meats have high contents of saturated (SFA) (465–508 g/kg of total fatty acids) and monounsaturated (MUFA) (439–456 g/kg of total fatty acids) fatty acids and small amounts of polyunsaturated fatty acids (PUFA) (43.7–83.1 g/kg of total fatty acids) [5,6] due to the ruminal biohydrogenation of unsaturated fatty acids into saturated fatty acids [7].

Recommendations on dietary levels of total lipids, cholesterol, SFA and trans-fatty acids are available from several references [8,9]. In general, meats that have high levels of PUFA (118.2–122.6 g/kg of total fatty acids) and low n-6/n-3 PUFA ratio (<4.0), combined with low cholesterol, are beneficial to human health and minimize cardiovascular disease [10,11]. Saturated fatty acids, particularly myristic (14:0) and palmitic (16:0) acids, can contribute to the increase of blood LDL in humans, and as a consequence, trigger coronary heart disease [12]. Moreover, MUFA and PUFA provide several benefits to human health [13].

The fatty acid composition and cholesterol content of beef are affected by multiple factors [14,15,16], diet being one of them. It has been shown that beef from animals fed mainly on pasture-based diets have an increase of about 60% of n-3 PUFA content in the meat, which leads to a more favorable n-6/n-3 PUFA ratio [7,17,18,19]. These findings putatively suggest that differences in fatty acid, cholesterol, tocopherol, tocotrienol and β-carotene contents may occur between water buffalo meat reared in grazing or feedlot systems.

In this context, and given the low extent to which water buffalo meat has been studied [20,21], the present study aimed to determine if the different production systems of Brazil Eastern Amazon influence the lipid composition of water Buffalo meat, assessed through total lipids, fatty acid composition, total cholesterol and lipid-soluble antioxidant vitamins (tocopherols, tocotrienols and β-carotene), and also evaluating the effect of seasonal variations (dry vs rainy periods).

## 2. Materials and Methods

### 2.1. Animals and Sampling

The Murrah × Mediterranean crossbred buffaloes were reared in four different production systems in commercial farms (see Section 2.2) in Brazil. A total of seven management systems were studied (three extensive production systems + one feedlot production system) as detailed in the subsequent section. All animals originated from dedicated meat production herds in Brazil (Figure 1).

After slaughter in a commercial abattoir following commercial practices, ribeye muscle samples (see Section 2.4) from twelve different animals per system were obtained. Each sampled farm was considered as representative of the production system for that region. This research received the approval waiver of the ethics committee on the use of animals (CEUA—Comissão de Ética no Uso de Animais) of the Universidade Federal Rural da Amazônia (protocol number: 4542190820).

In the three extensive production systems, 24 to 36-month-old water buffaloes were slaughtered with an averaged final body weight of 432 kg for the rainy season, and 409 kg for the dry season. In the feedlot production, 18-month-old water buffaloes were slaughtered with an averaged final body weight of 433 kg; all in commercial slaughterhouses. In all cases, the LL muscle was sampled, frozen and kept at −20 °C until further analysis.

### 2.2. Production Systems

System 1: Marajó Island—farm located in the Mesoregion of Marajó, Soure, Pará, Brazil (latitude 0°39′27.89″ S, longitude 48°42′35.01″ W, altitude 7 m). The pasture area was 224 ha with 374 animals and a stocking rate of 1.67 AU/ha. Tropical rainy climate Am, according to the Köpen classification, with average annual temperature of 27 °C, relative humidity (RH) of 85% and annual rainfall of 2500 mm. Two distinct seasonal periods were considered: rainy, from January to June (average temperature—AT of 27.7 °C, maximum temperature—MaxT of 31.7 °C, minimum temperature—MinT of 22.7 °C, RH of 87.2% and average rainfall—AR of 316.4 mm), and dry, from September to November (AT of 28.8 °C, MaxT of 32.9 °C, MinT of 23.3 °C, RU of 83.0% and AR of 122.2 mm) [22]. Buffaloes were raised in a traditional system and were exclusively fed on pasture composed of native grasses, such as capim-camalote (*Panicum elephantipes)*, Capim-andrequicé (*Leersia hexandra)* and canarana *(Hymenachne amplexicaulis)*, which were grown on floodable lands.

System 2: Santarém—farm located in the Mesoregion of the Lower Amazon, Santarém, Pará, Brazil (latitude 02°41′48.83″ S, longitude 54°38′35.43″ W, altitude 108 m). The pasture area was 483 hectares with 473 cattle and 267 buffaloes, and a stocking rate of 1.53 AU/ha. Tropical rainy climate Am, according to the Köpen classification, with an average annual temperature of 26 °C, relative humidity of 86% and rainfall of 2000 mm/year. The rainy period occurs between January and May (AT of 26.1 °C, MaxT of 29.8 °C, MinT of 23.8 °C, RH of 87.6% and AR of 296 mm), and the dry period takes place between July and November (AT of 27.2 °C, MaxT of 31.4 °C, MinT of 24.2 °C, RH of 84.4% and AR of 20 mm) [22]. Buffaloes were raised in a traditional system and were fed on native pastures, as in production system 1. However, these animals were also fed on implemented pastures, such as capim-mombaça (*Panicum maximum* cv. Mombaça) and braquiária (*Brachiaria brizantha*).

System 3: Nova Timboteua—farm located in the Northeastern Mesoregion of Pará, Nova Timboteua, Pará, Brazil (latitude 01°12′52.63″ S, longitude 47°24′30.94″ W, altitude 53 m). The pasture area was 825 hectares with 2000 animals, and a stocking rate of 2.42 AU/ha. Climate of Am type, according to Köpen classification, average annual temperature of 26.1 °C, relative humidity of 86% and annual precipitation of 2467 mm, with a dry period occurring from September to November (AT of 29.1 °C, MaxT of 33.5 °C, MinT of 22.0 °C, RU of 75.2% and AR of 25.1 mm) and a rainy period from December to August (AT of 27.2 °C, MaxT of 31.9 °C, MinT of 22.5 °C, RU of 86.1% and AR of 259.7 mm) [22]. The animals were fed pastures cultivated on dry forest lands (capim-mombaça and quicuio-da-amazônia—*Brachiaria humidicola*) and, in the dry season, supplemented with wet brewers’ grains residue (1 kg/animal/day). It is a by-product from the brewery industry that is extracted from the residue of barley malt with other cereal grains or grain products resulting from beer production.

Production system 4: Confinement (feedlot)—farm located in the Northeastern Mesoregion of Pará, Tomé-Açu, Pará, Brazil (latitude 02°25′08″ S, longitude 48°09′08″ W, altitude 45 m). The animals were housed in a feedlot at 12 months of age. The diet consisted of sorghum silage, soybean meal, wet sorghum premix and commercial feed, the latter being provided at 1 kg/animal (for high performance score).

### 2.3. Diet Sampling and Chemical Analysis

Pasture samples were collected in five different points (1 m^2^) from the three extensive production system farms. Samples were then homogenized, weighed and about 1 kg of sample stored at −20 °C until further analysis. Additionally, samples of the diet offered to the animals raised in the intense system were collected. All chemical analyses of pasture and feeds were performed at the Animal Nutrition Laboratory, Federal University of Pará/ Castanhal Campus, Pará, Brazil.

The diet samples were subjected to partial drying in a forced ventilation oven, at 60 °C for 24 to 72 h, to avoid losses of volatile compounds and chemical changes. Afterwards, the samples were left to cool at room temperature in order to minimize changes in humidity, and were ground in a 1 mm Willey mill for chemical analysis. Diets were analyzed for dry matter (DM; INCT-CA method G-003/1) at 105 °C for 16 h, and ash content (MM; INCT-CA M-001/1) in a muffle at 600 °C for 4 h.

Total nitrogen (N) (INCT-CA method N-001/1) was quantified using a three-step micro Kjeldhal procedure (sulfuric acid digestion, basic distillation and hydrochloric acid titration). Crude protein (CP) was determined as N × 6.25.

The neutral detergent fiber (DNF; INCT-CA F-001/1 and INCT-CA F-002/1) and acid detergent fiber (ADF; INCT-AC F-003/1 and INCT-AC F-004/1) contents were evaluated and corrected for protein and ash determinations, according to the methods recommended by the National Institute of Science and Technology in Animal Science (INCT-CA; [23]).

For non-fibrous carbohydrates determination, the procedure described by Sniffen et al. [24] was applied, whereas total digestible nutrients (TDN) were determined according to Clemson University equation: NDT = 93.59 − (ADF × 0.936).

### 2.4. Muscle Tissue Sampling

Twelve LL (ribeye) muscle samples were collected per group (production system and season). Sampling was taken after slaughter, when hides were removed and before meat maturation. Approximately 50 g (wet weight) of tissue were collected per sample and stored at −80 °C until further analysis. Then, muscle samples were lyophilized for approximately 48 h, until constant weight, using a Christ Alpha 1–2 LDplus freeze dryer (Christ alpha, Benningen, Germany).

### 2.5. Determination of Cholesterol, β-Carotene and Vitamin E

A simultaneous analysis of meat total cholesterol, β-carotene and vitamin E homologues was performed according to the methodology defined in detail by Prates et al. [25]. Briefly, vitamin E, β-carotene and cholesterol from LL meat samples were submitted to direct saponification and single extraction with *n*-hexane, in duplicate. Afterwards, organic phases were filtered through a 0.45 µm hydrophobic filter and analyzed by High-Performance Liquid Chromatography (HPLC) using a normal-phase silica column (Zorbax Rx-Sil column, 5 µm particle diameter, 4.6 mm ID × 25 cm, Agilent Technologies Inc. Santa Clara, CA, USA) and two serial detectors (diode array and fluorescence). Cholesterol, β-carotene and vitamin E homologues were quantified based on the external standard method using the calibration curves of peak area versus concentration.

### 2.6. Determination by HPLC in Normal Phase

The HPLC was used at a temperature of 20 °C. The mobile phase was hexane-isopropanol (99:1) (Flow: 1.0 mL/min) with an injection volume of 20 µL for α-tocopherol and 100 µL for the remaining tocopherols, and 20 or 100 µL for cholesterol. The detection was to tocopherols (e.g., 295 nm, em. 325 nm, PMT-Gain 14), cholesterol (202 nm); β-carotene (450 nm) and all-trans-retinol (325 nm).

### 2.7. Calculations and Control of Methods

Cholesterol, β-carotene and tocopherols were quantified by the external standard method. For tocopherol profile (α-tocopherol, β-tocopherol, γ-tocopherol and ∆-tocopherol), four calibration curves were used, one for cholesterol and another for β-carotene, respectively, of areas versus concentration.

### 2.8. Lipid Extraction and Fatty Acid Methylation and Analysis

Total lipids were extracted, in duplicate, from lyophilized LL meat samples as described by Folch et al. [26], with modifications of Carlson [27], and determined gravimetrically by weighing the lipid residue after solvent evaporation. Fatty acid methyl esters (FAME) were obtained, as previously described [28], after a combined transesterification, first in alkaline followed by acidic conditions. Then, FAME were extracted with n-hexane and analyzed by gas chromatography equipped with a flame ionization detector (GC-FID) (HP 7890A chromatography; Hewlett-Packard, Avondale, PA, USA) and separated using a fused capillary column (Supelcowax^®^ 10, 30 m, 0.2 mm id, film thickness of 0.20 mm; Supelco, Bellefonte, PA, USA). The chromatography conditions were previously reported [29]. Nonadecanoic acid methyl ester (19:0; Sigma, San Luis, EUA) was added as an internal standard. Fatty acid identification was obtained by comparison with a commercial standard FAME mixture of 37 components (Supelco, Inc., Bellefonte, PA, USA). Fatty acids were expressed as g/100 g of total fatty acids.

### 2.9. Nutritional and Health Lipid Indices

Fatty acid profile was used to determine nutritional and health lipid indices of buffalo LL meat, calculated according to the following equations [30,31,32]:$$n\text{-}6/n\text{-}3\mathrm{PUFA}=\frac{18:2\;n\text{-}6\;+\;18:3\;n\text{-}6\;+\;20:2\;n\text{-}6\;+\;20:3\;n\text{-}6\;+\;20:4\;n\text{-}6\;+\;22:4\;n\text{-}6}{18:3\;n\text{-}3\;+\;20:5\;n\text{-}3\;+\;22:5\;n\text{-}3\;+\;22:6\;n\text{-}3}\;$$
$$\mathrm{PUFA}/\mathrm{SFA}=\;\frac{\mathrm{PUFA}n\text{-}6\;+\mathrm{PUFA}n\text{-}3}{4:0\;+\;6:0\;+\;8:0\;+\;10:0\;+\;12:0\;+\;14:0\;+\;15:0\;+\;16:0\;+\;17:0\;+\;18:0\;+\;20:0}$$
$$h/H=\frac{18:1\;c9\;+\mathrm{PUFA}n\text{-}6\;+\mathrm{PUFA}n\text{-}3}{12:0\;+\;14:0\;+\;16:0}$$
IA (index of atherogenicity) = [(12:0) + (4 × 14:0) + (16:0)] [(ΣPUFA *n*-6 + ΣPUFA *n*-3) + ΣMUFA)]
IT (index of thrombogenicity) = [(14:0) + (16:0) + (18:0)]/[(0.5 × ΣMUFA) + (0.5 × ΣPUFA *n*-6) + (3 × ΣPUFA *n*-3) + (ΣPUFA *n*-3 × ΣPUFA *n*-6)]

### 2.10. Statistical Analysis

The experimental design was a completely randomized 3 × 2 + 1 factorial arrangement (three production systems, two seasonal periods and an additional treatment corresponding to the confinement in feedlot). The variables were analyzed in the PROC MIXED of SAS version 9.1 (2014) (SAS Institute Inc., Cary, NC, USA), considering the following model: yijk = µ + αi + βj + γij + ԑijk and yh = µ + δα + Ԑh, where:

yijk: variable response related to the level of the first factor (i = 1, 2 and 3) combined with the level of the second factor (j = 1, 2) in a specific repetition (k = 1, 2, … r);

µ: general mean;

αi: effect of the first factor level (Local—production systems) (i);

βj: effect of the second factor level (seasonal periods) (j);

γij: effect of the interaction of first factor level (i) with second factor level (j);

ԑijk: experimental error associated with observation yijk and assumption that ԑĳk~N (0, σ2) with an independent structure;

yh: variable response associated with the observation (h = 1.2, … m) of additional treatment (confinement in feedlot).

δα: effect of the additional treatment.

Ԑh: experimental error associated with additional treatment with the assumption that Ԑh~N (0, σ2) with an independent structure.

Production systems (extensive and intensive) and seasonal period (dry and rainy season) were considered as fixed effects within the model. Contrasts between extensive and intensive production systems, as well as interaction between experimental location and period of the year, were considered. The Tukey-Kramer post-hoc test was applied since data were unbalanced due to losses. The level significance of α = 0.05 was assumed to assess differences between least-square means of each group.

The choice of the statistical analysis aimed to preserve one of the most interesting aspect of our research: the differences between the several existing extensive production systems. This aspect is particularly relevant given the fact that production and field conditions of the extensive systems are considerably different between each other. Furthermore they are located in ecological regions with significant differences, furthermore located in an extremely large geographical area in the Brazilian Amazon. Indeed, each extensive system is representative of a different production reality and region.

## 3. Results and Discussion

### 3.1. Diet Composition

Table 1 shows the chemical composition of the diets fed to crossbred buffaloes raised extensively in three types of production systems during the dry and rainy seasons or intensively in feedlots.

The chemical composition of pasture among the three extensive production systems and the two seasonal periods did not show major differences. However, fatty acids such as lauric (12:0), myristic (14:0) and myristoleic (14:1c9) had the lowest values in the intensive system and brewers’ grains residue diets, and palmitic acid (16:0) presented the highest values in diets from the rainy season of Marajó and Nova Timboteua.

The lowest value of total lipids in the meat of buffaloes raised in Marajó island during the rainy season was likely due to the lower amount of total lipids (4.04 mg/g diet, Table 1) present in the plant mass ingested by such animals. Interestingly, the total lipid content of buffaloes from extensive systems is similar to that of animals from intensive systems, differing from the results found by Maniaci et al. [33] who observed a higher lipid content in the meat of stabled young bulls than those raised on pasture. Probably, the homogeneous slaughter weight between the two production systems (feedlot and pasture) in the present study may have neutralized the different lipid levels of the diet (Table 1). Regarding animals produced in extensive systems, there was only a residual effect of the location of production on meat total lipids with no seasonal variation (Table 2).

### 3.2. Fat Soluble Compound Muscle Composition

The fat-soluble compounds composition of ribeye meat of buffaloes extensively reared in three types of production systems in Amazonia, during dry and rainy seasonal periods, or intensively reared in feedlot are shown in Table 2.

The lowest value of total lipids (10.9 mg/g meat) was found in the meat of buffaloes raised on Marajó island during the rainy season (Table 2).

The highest values of cholesterol found in the meat of buffaloes raised in Santarém during the dry season (0.54 mg/g of meat), in Nova Timboteua during the rainy season (0.56 mg/g of meat) and in feedlot (0.53 mg/g of meat) (Table 2) are higher than those reported by Calabro et al. (2014) for confined Mediterranean buffaloes (0.32 mg/g meat). However, Giuffrida-Mendoza et al. [20] described a similar value in 19-month-old buffaloes (0.53 mg/g meat). On the other hand, animals raised on Marajó Island, regardless of the period of the year, had lower cholesterol value (average of 0.35 mg/g meat). A possible explanation for such results could be related to the pasture quality in this particular region that could have cholesterol and lipid-lowering effects. These levels may be correlated with the presence of MUFA in cis configuration, which was shown to reduce cholesterol levels in human plasma without reducing high-density lipoproteins (HDL), and, thus, possibly acts as a protective agent against coronary heart disease as previously suggested [34,35,36,37]. A cholesterol intake of less than 300 mg/day is recommended for healthy adults, and less than 200 mg/day for those with high levels of LDL cholesterol [9]. Thus, consuming 150 g of lean buffalo meat would provide less than 50% of the maximum amount of cholesterol allowed in the adult diet [9]. Even with the benefits of red meat consumption, the National Cancer Institute—INCA [38] recommends that the consumption of cooked red meat per week should not exceed 500 g.

The highest levels of α-tocopherol, 5.25 and 5.38 µg/g of meat, respectively, in buffaloes raised on the island of Marajó and in Nova Timboteua during the rainy season (Table 2) are consistent with the α-tocopherol values found in the corresponding diets (Table 1). The highest levels of γ-tocopherol (0.54 µg/g meat) and γ-tocotrienol (0.39 µg/g meat) observed in feedlot buffaloes, as well as in the rainy seasonal period of Nova Timboteua (0.23 µg/g meat), can be explained by the higher amount of these vitamins in the diet (Table 1). Although the levels of α-tocopherol and γ-tocopherol were higher in the pasture from Santarém during the dry period (Table 1), lower values of these vitamins were observed in the meat of these animals. This fact was likely due to the difficulty in controlling the actual animal feed intake, due to absence of fences and grazing control, especially in the dry seasonal period when there is low availability of grasses and legumes and those available have reduced nutritional value. Therefore, it can be inferred that the plant mass collected during this period may not fully represent the feed consumed.

### 3.3. Fatty Acid Muscle Composition

The fatty acid composition of ribeye meat of buffaloes extensively reared in three types of production systems in Amazonia, during dry and rainy seasonal periods, is shown in Table 3 and Table 4.

There was an interaction of production system and periods of the year (*p* < 0.05) for almost all the evaluated fatty acids. It is important to emphasize that SFA are related to a higher amount of LDL in the blood circulation leading to hypercholesteremia, which increases the concentration of lipids in the blood. On the other hand, MUFA, such as 18:1c9, and PUFA, like 18:2n-6, 18:3n-3 and 18:3n-6, increase the number of hepatic LDL receptors and, thus decrease LDL production, which consequently reduces circulating LDL and contributes to the improvement of human health, with a reduction of cardiovascular diseases [39,40]. In this study, the lowest percentages of 14:0 (0.52%) and 16:0 (15.68%) were determined in the meat of animals raised during the rainy season of Marajó, which demonstrates that buffaloes slaughtered at this time of the year have a healthier meat in relation to other production systems.

Among MUFA, 14:1c9 was found in higher percentages in all extensive systems and seasonal periods, whereas the percentages of 16:1c9 and 18:1c9 were higher in the meat of feedlot animals. The proportion of 17:1c9 was higher in the meat of animals raised during the rainy season in Santarém, which did not differ from the dry season in that location or from the rainy season in Marajó and Nova Timboteua. However, the main MUFA in beef is oleic acid (18:1c9), as it represents 89% of MUFA. This fatty acid is of high interest as it has a hypocholesterolaemic action, with the great advantage of not reducing HDL, and thus acting in the protection against coronary diseases [41].

The highest content of some SFA, such as margaric (17:0) and stearic (18:0) acids, in addition to the MUFA 17:1c9 and 18:1, were observed in the meat of animals raised in the feedlot system. The highest value of oleic acid (18:1c9) was found in the meat of animals raised in feedlot (39.1%) when compared to other treatments.

The 18:2c9,t11 was observed at the highest percentage (1.03%) in animals raised in the production system of Marajó Island, during the rainy season, but the lowest value (0.52%) was detected in animals reared in feedlots. These results indicate potential human health benefits of meat produced on Marajó Island in the rainy season conversely to buffalo meat from feedlots, as CLA was shown to have anticarcinogenic effects [42,43,44].

In the present study, the highest value of linoleic acid (18:2n-6) was found in buffalo meat (11.4%) raised in Santarém during the dry season, which was similar to the other treatments with the exception of the meat of animals raised in Marajó during the dry period that presented a lower value of this fatty acid (7.82%), probably due to poor nutritive plant mass diet caused by lack of rain and, thus, low feed intake.

The highest value of C18:3n-6 (0.17%) was found in the meat of buffaloes raised during the rainy season in Marajó or in feedlots, although this value did not differ from that found in the meat of buffaloes raised during the dry period in Santarém or both seasonal periods in Nova Timboteua. The n-6 PUFA series play a key role in brain development, cell lifespan and structure and skin protection [45,46,47]. However, if consumed in excess, they can have negative effects, such as premature cell aging, changes in cell membranes, abnormalities in cell multiplication or carcinoma induction [45,46,47,48].

In the present study, the highest percentage of 18:3n-3 was observed in the meat of animals raised during the rainy season in Marajó (2.88%), similarly to what was found for the rainy season of Nova Timboteua and the dry season of Santarém (Table 2). The n-3 fatty acids were identified as having potential health beneficial activities, such as the prevention of cardiovascular diseases, stimulation of cerebral cortex development and cognitive capacity of children [45,47,49].

The sum of MUFA was higher in the meat of feedlot animals, with 44.9%. The sum of PUFA was significantly higher in the meat of buffaloes raised in extensive production systems, particularly during the dry season of Santarém (25.2%) or the rainy season of Marajó island (26.3%) and Nova Timboteua (23.3%), compared to intensive systems (15.3%). The sum of n-3 fatty acids was higher in the meat of animals raised during the rainy season of Marajó (8.62%), but that did not differ from the values found in the dry season of Marajó and Santarém, and in the rainy season of New Timboteua.

The sum of n-3 PUFA was significantly different among the production systems studied, with lower values in the meat of animals reared in feedlots, which was due to the higher and lower amounts of cereals and plant mass, respectively, in this diet [50]. These results might indicate a lower bioactivity of meat from feedlots compared to extensive systems, considering the n-3 PUFA bioactive properties that include anti-inflammation properties [51,52,53,54,55].

The highest percentages of 18:2c9,t11, 20:2n-6, 20:4n-6, 20:5n-3 (EPA), 22:5n-3, 22:6n-3 (DHA), ΣPUFA and Σn-3 PUFA were observed in the meat of buffaloes raised during the rainy season in Marajó island, which was probably due to the fact that this system provided a higher availability of plant mass in pastures than others. These essential fatty acids should be included in the diet of humans and other mammals [49], as they are important for maintaining and promoting health. Alabiso et al. [56] found higher content of PUFA’s on bresaola (a product made with the *Semimembranosus*, *Semitendinosus* and *Biceps brachii* muscles) of young bulls fed a pasture-based diet. It was similar to what we observed in this study, with most PUFA’s from the meat of animals raised in the pasture (mainly in the Marajó island).

The highest proportions of DMA 18:0 (1.26%) and ΣDMA (2.48%) were found in the meat of buffaloes raised during the rainy season in Nova Timboteua. For ΣPUFA, the highest value (25.2%) was observed in the meat of animals raised during the dry period in Santarém, which may have occurred due to the better quality and higher availability of plant mass for the dry season of the Lower Amazon. It is important to emphasize that during the rainy season in Santarém, Lower Amazon, the pastures are completely flooded by the rivers, which makes it difficult for the animals to graze. In addition, during the dry period, when the water level decreases, the floodplains are rich in nutrients from the sediments formed at the time of flooding, therefore providing pastures with better nutritional value to the animals. However, the values of ΣPUFA present in the meat of buffaloes raised on the island of Marajó (26.3%) and in Nova Timboteua (23.3%) were higher than for the other production systems and similar to each other, both during the rainy season that is a period of higher availability of plant mass in the pastures.

The sum of hypocholesterolaemic fatty acids (Σh) was higher in the meat of animals raised in feedlots (54.5%), although it did not significantly differ from the values found in all pasture-based production systems and periods of the year, except for the value found in the dry seasonal period of Nova Timboteua, which was significantly lower (49.5%). The sum of hypercholesterolaemic fatty acids (ΣH) was higher in the meat of buffaloes raised in feedlot (20.7%), which differed (*p* < 0.05) from the ΣH values obtained for extensive production systems. However, it did not differ significantly from the values found in the rainy season of Santarém or in the dry season of Nova Timboteua and Marajó Island.

The PUFA/SFA ratio value was higher in the meat of animals raised during the dry season in Santarém (0.66) and the rainy season in Marajó Island (0.70). These values are consistent with the higher availability of pastures in the two production systems. The PUFA/SFA values found in the meat of the animals in the studied production systems were higher than 0.45, except in the confinement, with a value of 0.42. It is important to highlight that the recommended nutritional level of the PUFA/SFA ratio in the human diet must be higher than 0.45 [30].

Although extensive production systems showed significant differences for the n-6/n-3 PUFA ratio present in buffalo meat, the values were lower than 4, which is in accordance with previous epidemiological studies that recommended a n-6/n-3 ratio lower than 4 in the human diet [30]. Conversely, the intensive production system (feedlot) led to a higher ratio (6.15). The latter value was due to the presence of concentrates and forage sorghum in the diet of confined buffaloes. Animals fed exclusively with grains can have a n-6/n-3 ratio value of about 15 in the meat [31,50].

The n-6/n-3 PUFA ratio is particularly beneficial in the meat from ruminants, especially in animals that have consumed plant mass [40,50]. This occurred in buffaloes raised in extensive systems in the present study, which presented high levels of α-linolenic acid (18:3n-3), in contrast to confined animals (Table 2). Similarly, Luo et al. [55] found high levels of C18:3n3 in the meat of small ruminants fed on pasture grass. It is important to emphasize that a high n-6/n3 ratio is considered a risk factor in the development of some types of cancer and cardiovascular diseases. The increased consumption of foods that supply n-3 fatty acids reduces the incidence of these diseases [45,51,52,53]. Likewise, excess n-6 fatty acids can interfere with the metabolism of n-3 fatty acids, which alters their biological effects [48].

In the present study, the values of h/H ratio were above 2 and were influenced by the seasonal period. For an evaluation of the functional effects of fatty acids, the ratio between fatty acids with hypocholesterolaemic and hypercholesterolaemic effects was established as a way of evaluating the health benefits of meat. For meat products, the value of 2 for h/H ratio was referenced. Values higher than 2 correspond to meats of superior nutritional quality with an abundance of fatty acids that promote the reduction of plasma cholesterol (hypocholesterolaemic), and thus a decreased risk of cardiovascular diseases [57,58,59,60,61].

The highest values of fatty acid IA were found in animals raised in Marajó (0.38) and Nova Timboteua (0.39) during the dry seasonal period, and in feedlot animals (0.39). On the other hand, the lowest IA was found in the meat of animals raised extensively in Marajó Island during the rainy season (0.31%). The lower levels of fatty acids IT were observed in the meat of buffaloes raised in the different extensive production systems.

The IT considers the fatty acids 14:0, 16:0 and 18:0 as thrombogenic, and the MUFA and PUFA (n-6 and n-3) as anti-thrombogenic [31,45]. There are no recommended values for IT and IA, although lower values indicate a more promising fatty acid ratio for human health promotion. Consequently, a meat with low IT and IA values might be considered of better nutritional quality for humans, as it presents a higher amount of anti-atherogenic fatty acids than meats with high ratios, possibly contributing to coronary heart disease prevention [62,63].

In general, we can infer that meat from buffaloes raised on pasture, especially in the rainy season, would bring greater benefits to human health concerning animals raised in confinement. We confirmed it by the best values of mono and polyunsaturated fatty acids in addition to other attributes, such as lipids and vitamins. The hypothesis is that these results were obtained due to the feeding to which these animals were submitted and that better pasture quality in the rainy season provided beneficial effects on the meat nutrient composition.

## 4. Conclusions

It is concluded that the different Amazon production systems and seasonal periods affected meat composition regarding cholesterol, tocopherols and fatty acids from water buffalos. Among all production systems studied, the buffalo meat reared on the Marajó island during the rainy season seems to be the healthier option for human consumption as it has less cholesterol, more α-tocopherol, less saturated fatty acids (14:0 and 16:0) and higher amounts of MUFA and PUFA (18:1; 18:3 n-6; 18:3n-3; 18:2c9,t11; 20:2n-6; 20:4n-6; 20:5n-3; 22:5n-3 and 22:6n-3). In addition, this meat seems to have higher n-3 PUFA percentages and PUFA/SFA and h/H ratios, and lower n-6/n-3 PUFA ratios and AI values. However, meat from all extensive production systems showed similar and relatively low values of IT, which is beneficial for human health. Then we found that animals raised on pasture, especially in the rainy season, may have better nutritional characteristics of meat lipids than those raised in confinement.

## Figures and Tables

**Figure 1 animals-12-00595-f001:**
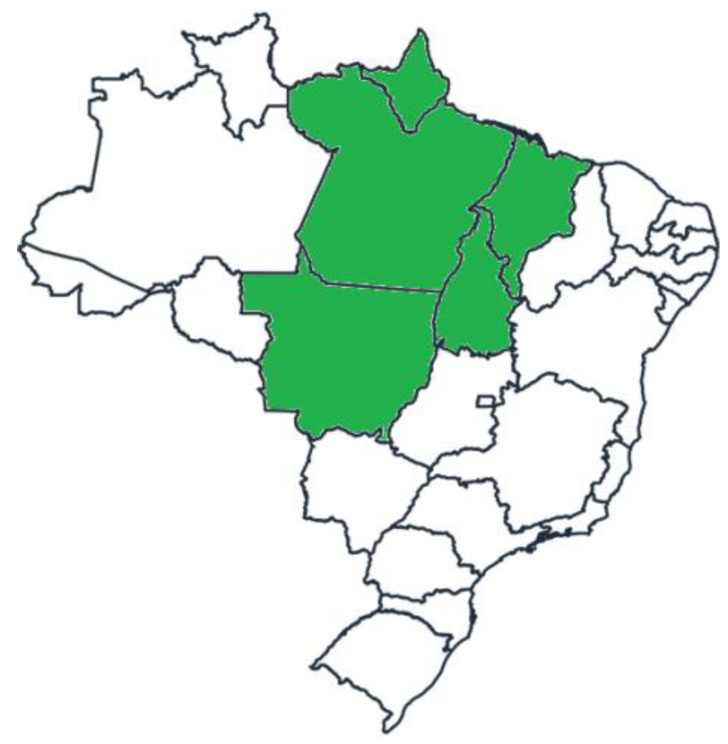
The Eastern Amazon of Brazil, encompassing the states of Amapá (AP), Pará (PA), Maranhão (MA), Tocantins (TO) and Mato Grosso (MT).

**Table 1 animals-12-00595-t001:** Chemical composition of diets.

	Santarém		Marajó		Nova Timboteua			Intensive System
Compounds	DS	RS	DS	RS	DS		RS	
Chemical composition (% DM)					Plant mass	Brewers grains residue		
DM	23.95	24.10	23.87	18.31	23.32	26.12	24.71	39.00
OM	91.53	90.98	89.45	84.86	91.79	96.88	95.35	94.77
CP	7.86	8.72	7.56	8.86	7.73	28.03	9.38	8.27
NDF	73.23	79.07	68.72	70.91	75.73	55.69	68.96	54.36
NFC	9.07	1.79	11.79	3.09	6.27	5.16	13.85	29.58
ADF	44.87	54.90	40.05	43.90	55.80	22.55	44.51	38.08
TDN	51.59	42.20	56.09	52.5	41.35	72.48	51.93	57.94
Ash	8.47	9.02	10.55	15.14	8.21	3.12	6.17	5.23
TL (mg/g diet)	4.84	4.71	5.17	4.04	5.07	-	2.88	9.77
α-Tocopherol (µg/g DM)	7.76	4.10	5.33	6.77	6.75	23.7	9.40	17.00
γ-Tocopherol (µg/g DM)	0.90	0.12	0.13	0.15	0.16	3.27	0.44	0.59
γ-Tocotrienol (µg/g DM)	0.71	0.98	1.65	2.00	1.90	27.54	2.16	nd
Fatty acid composition (% /total FA)								
12:0	3.03	1.85	1.84	1.03	2.45	0.03	1.03	0.11
14:0	3.57	2.81	3.35	3.33	2.63	0.38	3.33	0.18
14:1c9	0.86	1.73	2.27	1.41	0.38	0.00	1.41	0.07
15:0	1.05	0.96	0.76	0.99	0.55	0.11	0.99	0.13
16:0	37.52	40.82	36.60	50.63	32.61	25.30	50.65	18.50
16:1c9	2.11	1.96	1.28	1.09	0.72	0.18	1.09	0.34
17:0	3.87	3.27	6.38	4.26	2.30	0.15	4.26	0.44
17:1c9	1.52	1.23	0.66	2.05	0.29	0.04	2.05	0.07
18:0	11.77	6.93	13.93	9.61	6.63	1.52	9.61	2.59
18:1c9	12.49	7.70	11.33	5.45	20.53	9.89	5.45	33.63
18:1c11	1.96	1.54	2.10	1.60	1.82	0.85	1.60	1.77
18:2n-6	10.35	14.25	9.45	7.37	21.17	55.25	7.37	38.18
18:3n-6	0.00	0.00	0.61	0.00	0.00	0.00	0.00	0.00
18:3n-3	5.71	12.71	4.01	7.37	5.81	5.56	7.37	3.38
20:0	3.92	2.23	5.43	3.81	1.66	0.25	3.81	0.41
20:1c11	0.00	0.00	0.00	0.00	0.44	0.50	0.00	0.22

Note: Diets (*n* = 3) fed to Murrah × Mediterranean crossbred buffaloes that were extensively (three production system types × dry (DS) and rainy (RS) seasonal periods) or intensely reared. nd, not detected; DM, dry matter; OM, organic matter; CP, crude protein; NDF, neutral detergent fiber; NFC, non-fiber carbohydrates; ADF, acid detergent fiber; TDN, total digestible nutrients; TL, total lipids.

**Table 2 animals-12-00595-t002:** Fat-soluble compounds of ribeye muscle of crossbred buffaloes (*n* = 12) raised extensively in three types of production systems, during dry (DS) and rainy (RS) seasonal periods, or intensely in feedlot.

	Extensive Systems								*p*-Value			
	Santarém (S)		Marajó (M)		Nova Timboteua (N)		Intensive System	SEM	Extensive vs. Intensive	Local (L) 1	Period (P) 2	L × P
Compounds	DS	RS	DS	RS	DS	RS						
TL (mg/g muscle)	12.71 ^ab^	13.22 ^ab^	11.22 ^ab^	10.90 ^b^	14.81 ^a^	12.83 ^ab^	13.17 ^ab^	0.852	0.553	0.0100 (S = M < N)	0.418	0.391
TC (mg/g muscle)	0.54 ^a^	0.41 ^b^	0.34 ^b^	0.37 ^b^	0.36 ^b^	0.56 ^a^	0.53 ^a^	0.019	<0.001	<0.001	0.037	<0.001
α-Tocopherol (µg/g muscle)	1.18 ^d^	2.71b ^c^	3.44b ^c^	5.25 ^a^	2.20 ^cd^	5.38 ^a^	3.60 ^b^	0.295	0.457	<0.001	<0.001	0.032
γ-Tocopherol (µg/g muscle)	0.02 ^c^	0.18 ^b^	nd	nd	nd	0.16 ^b^	0.54 ^a^	-	-	0.001	<0.001	-
γ-Tocotrienol (µg/g muscle)	0.03 ^b^	0.09 ^b^	nd	nd	nd	0.23 ^a^	0.39 ^a^	-	-	0.074	0.002	-

Note: ribeye muscle of Murrah × Mediterranean crossbred buffaloes (*n* = 12) raised extensively in three types of production systems, during dry (DS) and rainy (RS) seasonal periods, or intensely in feedlot. TL, total lipids; TC, total cholesterol; ^1^ production systems; ^2^ season periods; (S = M < N) indicates the difference among the location (Santarém, Marajó and Nova Timboteua). ^a,b,c,d^ Values with different superscripts within a row differ significantly at *p <* 0.05. nd, not detected.

**Table 3 animals-12-00595-t003:** Fatty acid composition of ribeye muscle of crossbred buffaloes (*n* = 12) raised extensively in three types of production systems, during dry (DS) and rainy (RS) seasonal periods, or intensely in feedlots, in Eastern Amazonia (Brazil).

	Extensive Systems								*p*-Value			
	Santarém (S)		Marajó (M)		Nova Timboteua (N)		Intensive System	SEM	Extensive vs. Intensive	Local (L) ^1^	Period (P) ^2^	L × P
Compounds	DS	RS	DS	RS	DS	RS						
Fatty acid composition (% total fatty acids)												
12:0	0.40	0.29	0.34	0.39	0.29	0.41	0.23	0.029	0.000	0.694	0.468	0.002
14:0	0.78 ^ab^	0.89 ^a^	0.84 ^a^	0.52 ^b^	0.97 ^a^	0.79 ^ab^	1.01 ^a^	0.080	0.017	0.034	0.049	0.020
14:1c9	0.16 ^a^	0.10 ^ab^	0.11 ^ab^	0.11 ^ab^	0.14 ^ab^	0.07 ^b^	0.13 ^ab^	0.017	0.566	0.299	0.009 (DS > RS)	0.067
15:0	0.31 ^a^	0.31 ^a^	0.36 ^a^	0.35 ^a^	0.31 ^a^	0.33 ^a^	0.17 ^b^	0.019	<0.001	0.076	0.927	0.735
16:0	16.91 ^ab^	17.59 ^a^	17.56 ^a^	15.68 ^b^	17.64 ^a^	16.19 ^ab^	19.49 ^ab^	0.437	<0.001	0.393	0.022	0.015
16:1c9	1.57 ^c^	2.04 ^ab^	1.51 ^c^	1.75 ^bc^	1.60 ^c^	1.68 ^bc^	2.41 ^a^	0.091	<0.001	0.055	0.000 (DS < RS)	0.060
17:0	1.55 ^abc^	1.50 ^bc^	1.58 ^abc^	1.68 ^ab^	1.37 ^c^	1.78 ^a^	0.71 ^d^	0.051	<0.001	0.169	0.002	0.001
17:1c9	0.70 ^abc^	0.81 ^a^	0.68 ^bcd^	0.69 ^abc^	0.60 ^cd^	0.75 ^ab^	0.56 ^d^	0.030	<0.001	0.022 (S > M = N)	0.001 (DS < RS)	0.116
18:0	18.98 ^a^	17.78 ^ab^	19.49 ^a^	18.85 ^a^	18.98 ^a^	17.60 ^ab^	15.15 ^b^	0.656	<0.001	0.398	0.071	0.861
18:1^(1)^	3.33 ^b^	3.60 ^ab^	3.32 ^b^	4.01 ^a^	3.98 ^a^	3.58 ^ab^	2.56 ^c^	0.114	<0.001	0.044	0.071	0.000
18:1c9	25.55 ^c^	31.22 ^b^	29.73 ^bc^	25.66 ^c^	28.08 ^bc^	27.92 ^bc^	39.11 ^a^	1.053	<0.001	0.813	0.594	0.000
18:2n-6	11.44 ^a^	8.15 ^ab^	7.82 ^b^	10.54 ^ab^	11.06 ^ab^	10.10 ^ab^	8.12 ^ab^	0.755	0.038	0.213	0.432	0.001
18:3n-6	0.12 ^abc^	0.11 ^bc^	0.08 ^c^	0.17 ^a^	0.14 ^ab^	0.12 ^abc^	0.17 ^a^	0.013	0.001	0.494	0.031	<0.001
18:3n-3	2.67 ^ab^	2.09 ^bc^	1.85 ^c^	2.88 ^a^	1.85 ^c^	2.50 ^abc^	0.43 ^d^	0.170	<0.001	0.500	0.021	0.000
18:2c9,t11	0.89 ^ab^	0.76 ^b^	0.75 ^b^	1.03 ^a^	0.71 ^b^	0.86 ^ab^	0.52 ^c^	0.041	<0.001	0.050	0.007	<0.001
20:0	0.16 ^ab^	0.14 ^bc^	0.19 ^ab^	0.21 ^a^	0.19 ^ab^	0.15 ^abc^	0.09 ^c^	0.014	<0.001	0.008 (M > S = N)	0.397	0.122
20:1c11	0.08 ^b^	0.10 ^b^	0.09 ^b^	0.10 ^ab^	0.11 ^ab^	0.08 ^b^	0.14 ^a^	0.009	<0.001	0.699	0.613	0.049
20:2n-6	0.47 ^b^	0.43 ^b^	0.56 ^b^	0.83 ^a^	0.36 ^b^	0.50 ^b^	0.50 ^b^	0.049	0.614	<0.001	0.006	0.017
20:3n-6	1.18	0.92	1.06	1.08	1.05	1.06	0.92	0.100	0.214	0.969	0.376	0.295
20:4n-6	4.77 ^ab^	4.13 ^ab^	4.10 ^ab^	5.05 ^a^	3.66 ^ab^	4.44 ^ab^	3.27 ^b^	0.377	0.010	0.401	0.266	0.093
20:5n-3	1.93 ^ab^	1.36 ^bc^	2.33 ^a^	2.51 ^a^	1.17 ^bc^	1.93 ^ab^	0.64 ^c^	0.175	<0.001	<0.001	0.433	0.005
22:5n-3	2.15 ^ab^	1.74 ^bc^	2.12 ^ab^	2.52 ^a^	1.76 ^bc^	2.16 ^ab^	1.06 ^c^	0.169	<0.001	0.072	0.394	0.048
22:6n-3	0.49 ^b^	0.41 ^b^	0.49 ^b^	0.72 ^a^	0.39 ^b^	0.46 ^b^	0.22 ^c^	0.039	<0.001	0.000	0.037	0.002
DMA16:0	0.63	0.64	0.43	0.48	0.92	1.11	0.38	0.166	0.077	0.0010 (S = M < N)	0.550	0.868
DMA18:0	0.62 ^ab^	0.52 ^b^	0.35 ^b^	0.51 ^b^	0.87 ^ab^	1.26 ^a^	0.36 ^b^	0.158	0.058	0.002 (S = M < N)	0.279	0.367
DMA18:1	0.10	0.07	0.08	0.05	0.10	0.11	0.07	0.018	0.339	0.110	0.248	0.383
Other	2.97	3.09	2.93	2.66	2.43	2.92	2.11	0.322	0.041	0.609	0.702	0.5720

SEM, standard error of the mean; ^1^ production systems; ^2^ season periods; (S > M = N), (S = M < N) and (M > S = N) indicate the difference among the locations (Santarém, Marajó and Nova Timboteua); (DS > RS) and (DS < RS) indicate the difference between the periods (dry and rainy). 12:0, lauric; 14:0, myristic; 14:1c9, myristoleic; 15:0, pentadecylic; 16:0, palmitic; 16:1c9, palmitoleic; 17:0, margaric; 17:1c9, heptadecaenoic; 18:0, stearic; 18:1^(1)^ = sum of 18:1*cis* and 18:1*trans*; 8:1c9, oleic; 18:2n-6, linoleic acid; 18:3n-6, γ-linolenic; 18:3n-3, α-linolenic; 18:2c9,t11, conjugated linoleic—CLA; 20:0, arachidic; 20:1c11, eicosaenoic; 20:2n-6, eicosadienoic; 20:3n-6, dihomo-gammalinoleic; 20:4n-6, arachidonic; 20:5n-3, eicosapentaenoic; 22:5n-3, docosapentaenoic; 22:6n-3, docosahexaenoic; DMA, dimethyl acetals; other, non-identified fatty acids. ^a,b,c,d^ Values with different superscripts within a row differ significantly at *p <* 0.05.

**Table 4 animals-12-00595-t004:** Partial sums of fatty acids and fatty acid nutritional indices of ribeye muscle of crossbred buffaloes (*n* = 12) raised extensively in three types of production systems, during dry (DS) and rainy (RS) seasonal periods, or intensely in feedlot, in Eastern Amazonia (Brazil).

	Extensive Systems								*p*-Value			
	Santarém (S)		Marajó (M)		Nova Timboteua (N)		Intensive System	SEM	Extensive vs. Intensive	Local (L) ^1^	Period (P) ^2^	L × P
Compounds	DS	RS	DS	RS	DS	RS						
Partial sums of fatty acids (% total fatty acids)												
ΣSFA	39.09	38.47	40.36	37.68	39.75	37.24	36.85	0.887	0.051	0.867	0.019 (DS > RS)	0.503
ΣMUFA	31.40 ^c^	37.87 ^b^	35.44 ^bc^	32.33 ^c^	34.51 ^bc^	34.08 ^bc^	44.90 ^a^	1.140	<0.001	0.812	0.317	0.000
ΣPUFA	25.19 ^a^	19.33 ^ab^	20.40 ^ab^	26.29 ^a^	21.43 ^ab^	23.28 ^a^	15.33 ^b^	1.650	0.000	0.787	0.665	0.004
Σn-6	17.97	13.73	13.61	17.67	16.26	16.23	12.98	1.206	0.028	0.888	0.946	0.0053
Σn-3	7.23 ^ab^	5.60 ^b^	6.79 ^ab^	8.62 ^a^	5.17 ^b^	7.05 ^ab^	2.35 ^c^	0.514	<0.001	0.015	0.140	0.003
Σh	50.75 ^ab^	50.55 ^ab^	50.13 ^ab^	51.95 ^ab^	49.51 ^b^	51.20 ^ab^	54.45 ^a^	1.030	0.001	0.834	0.241	0.609
ΣH	18.08 ^bc^	18.76 ^ab^	18.75 ^ab^	16.59 ^bc^	18.90 ^ab^	17.39 ^bc^	20.73 ^a^	0.477	<0.001	0.319	0.020	0.017
ΣDMA	1.36 ^ab^	1.24 ^ab^	0.87 ^b^	1.05 ^ab^	1.88 ^ab^	2.48 ^a^	0.81 ^b^	0.330	0.065	0.004 (S = M < N)	0.444	0.606
Fatty acid nutritional indices												
PUFA/SFA	0.66 ^a^	0.51 ^ab^	0.52 ^ab^	0.70 ^a^	0.55 ^ab^	0.63 ^ab^	0.42 ^b^	-	-	0.9053	0.4734	0.0171
n-6/n-3	2.72 ^bc^	2.68 ^bc^	2.15 ^c^	2.21 ^c^	3.73 ^b^	2.45 ^c^	6.15 ^a^	0.250	<0.001	<0.001	0.004	0.000
h/H	2.83 ^ab^	2.73 ^ab^	2.72 ^ab^	3.15 ^a^	2.67 ^ab^	2.97 ^ab^	2.64 ^b^	0.116	0.105	0.434	0.048	0.092
IA	0.36 ^ab^	0.37 ^ab^	0.38 ^a^	0.31 ^b^	0.39 ^a^	0.35 ^ab^	0.39 ^a^	0.017	0.079	0.339	0.015	0.055
IT	0.24 ^b^	0.34 ^b^	0.32 ^b^	0.19 ^b^	0.37 ^b^	0.25 ^b^	0.56 ^a^	0.041	<0.001	0.486	0.161	0.010

SEM, standard error of the mean; ^1^ production systems; ^2^ season periods; (S = M < N) indicate the difference among the locations (Santarém, Marajó and Nova Timboteua); (DS > RS) indicates the difference between the periods (dry and rainy); SFA, saturated fatty acids; MUFA, monounsaturated fatty acids; PUFA, polyunsaturated fatty acids; ΣSFA = sum of 12:0, 14:0, 15:0, 16:0, 17:0, 18:0 and 20:0; ΣMUFA = sum of 14:1c9, 16:1c9, 17:1c9, 18:1c9, 18:1 and 20:1c11; ΣPUFA = sum of 18:2n-6, 18:3n-6, 18:3n-3, 20:2n-6, 20:3n-6, 20:4n-6, 20:5n-3, 22:5n-3 and 22:6n-3; Σn-6 = sum of 18:2n-6, 18:3n-6, 20:2n-6, 20:3n-6 and 20:4n-6; Σn-3 = sum of 18:3n-3, 20:5n-3, 22:5n-3 and 22:6n-3; PUFA/SFA = (Σ18:2n-6, 18:3n-6, 18:3n-3, 20:2n-6, 20:3n-6, 20:4n-6, 20:5n-3, 22:5n-3 and 22:6n-3)/(Σ12:0, 14:0, 15:0, 16:0, 17:0, 18:0 and 20:0); n-6/n-3 = (Σ18:2n-6, 18:3n-6, 20:2n-6, 20:3n-6 and 20:4n-6)/(Σ18:3n-3, 20:5n-3, 22:5n-3 and 22:6n-3); h/H = hypocholesterolaemic/hypercholesterolaemic ratio = (Σ18:1c9, 18:2n-6, 18:3n-6, 18:3n-3, 20:2n-6, 20:3n-6, 20:4n-6, 20:5n-3 and 22:5n-3)/(Σ12:0, 14:0 and 16:0); IA = index of atherogenic = [(12:0) + (4 × 14:0) + (16:0)]/[(Σ18:2n-6, 18:3n-6, 20:2n-6, 20:3n-6 e 20:4n-6 + 18:3n-3, 20:5n-3, 22:5n-3 and 22:6n-3) + (Σ14:1c9, 16:1c9, 17:1c9, 18:1c9, 18:1 and 20:1c11)]; IT = index of thrombogenic = [Σ14:0, 16:0 and 18:0)]/[0.5 × (14:1c9 + 16:1c9 + 17:1c9 + 18:1c9 + 18:1 + 20:1c11)] + [(0.5 × (18:2n-6 + 18:3n-6 + 20:2n-6 + 20:3n-6 + 20:4n-6)] + [3 × (18:3n-3 + 20:5n-3 + 22:5n-3 + 22:6n-3)] + (Σ18:3n-3, 20:5n-3, 22:5n-3 and 22:6n-3 × Σ18:2n-6, 18:3n-6, 20:2n-6, 20:3n-6 and 20:4n-6)]; ΣDMA = DMA 16:0, 18:0 and 18:1. ^a,b,c^ Values with different superscripts within a row differ significantly at *p* < 0.05.

## Data Availability

The data presented in this study are available upon reasonable request from the corresponding author.

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
