# Peer review of "Total Lipids, Fatty Acid Composition, Total Cholesterol and Lipid-Soluble Antioxidant Vitamins in the longissimus lumborum Muscle of Water Buffalo (Bubalus bubalis) from Different Production Systems of the Brazilian Eastern Amazon"

_animals, 2022, doi:10.3390/ani12050595_

Round 1

Reviewer 1 Report

The authors investigated the total lipids, fatty acid composition, total cholesterol and lipid-soluble antioxidant vitamins in the longissimus lumborum muscle of water buffalo (Bubalus bubalis) from different production systems of the Brazilian Eastern Amazon. They compared lipid components in 7 areas and systems. This manuscript (MS) was clearly written and easy to understand. However, some major issues significantly compromised the quality of this MS.

Comments

Abstract

  • Line 26, more aware of nutritional compositions of foods.
  • Line 26, please make sure you deleted the extra space between words and sentences.
  • Line 28, different production systems in the Eastern Amazon on meat nutritional quality parameters of water buffalo.
  • Line 32, please clarify pasture production system or intensive was better
  • Line 35, revise, seasonal variation is not a production system.
  • Line 38, in where?
  • Line 40, mention then in parentheses here.
  • Line 41, complete name or delete it from here (no abbreviations).
  • Line 44, complete name, what is h/H.
  • Line 46, complete name,
  • Please reorder the keywords alphabetically and capitalize each word.

Introduction:

  • Well-developed introduction and included a clear fellow and relevant points.
  • Line 63, revise it like the abstract
  • Line 64, please add the approx. values in parentheses.
  • Line 69, please shortly mention in parentheses.
  • Line 74, is not relative enough as buffalo meat has not enough n-3. Please revise or delete it.
  • Line 88, delete the point.
  • Lien 86, mentioning one time “Eastern Amazon” is enough and please write only Brazil for the rest of the MS.

Material and methods

  • Well-organized section. Clear fellow and all required details were provided.
  • Figure 1, please cover the name of the place in this map which can be consistent with the name of treatments.
  • Line 117, please mention somewhere in the MS the approx. temperature, rain and other information from dry and rainy seasons on the one table.
  • Line 120 and elsewhere, please first mention the common name plus scientific name, and for the rest of the MS, just report the common name.
  • The statistical analysis is so complicated. I suggest you use 3*2 two way ANOVA plus one independent sample T-Test (comparing intensive with an average of 6 extensive treatments. It will help to organize the results and discussion better.
  •  

Results

  • Well-written section, all necessary things have been covered but need to be more numeric.
  • Line 239, explanation about crossbred is for the M&M section.
  • Line 242, please revise it. You compare one time 6 treatments and then 7 treatments.
  • Line 251, please be consistent with the intensive system throughout the MS.
  • Table 1, you mentioned plant mass for Nova; what about Marajo and others?. Please be consistent with the provided information.
  • Line 253, please check my comment regarding the common name and scientific name.
  • In tables, you compared intensive and extensive (average of six treatments) and reported the P-value. Please add a column for the average of extensive, two seasons and locals. The rule of Two-way ANOVA is clear, which you did not consider. If the interaction is significant, you can unpack and compare 6 treatments with each other. If the interaction was not significant, you should compare the average of dry with rain and three locations. In this way, you can not compare the original data in the table with Tukey. Please see this paper as an example of how to unpack the data in two ways ANOVA. https://onlinelibrary.wiley.com/doi/10.1111/jpn.13659
  • Throughout the MS, if there is no significant difference, no need to report P-value.
  • Please update the result and discussion with the change of statistical analysis.
  • I will comment on Results when you update the MS with this change.
  •  
  • Discussion
  • Put the subheading for the discussion section like results. Also, keep a sequence in subheading for investigated factors, in M&M, result, and discussion.
    • You repeated the result in the discussion section. I suggest you combine the result and discussion sections or revise this section.
    • Line 362, please revise it.
    • Line 370, it is because of herbs they ate at this place had cholesterol and lipid-lowering effect. Please add this point. Also, you need to follow the rule of unpacking two-way ANOVA.
    • I will review the MS when you update the MS with these changes. Please send the revised version with track change as well to see how you improved the MS.
    • The discussion needs to be revised deeply and get better adjusted. For any parameters, please first discuss two wany ANOVA results between 6 treatments. Then, discuss intensive and extensive (average of 6 treatments) methods.
  • Some parts of the discussion are better updated with research in 2020 and 2021 as they refer to some old references. Please update the discussion with the latest studies as much as possible.
  • Although you wrote this section well, you can still improve it by answering these questions and annotating them to the discussion section. Why were these results observed? Discuss more possible reasons.
  • The conclusion needs to be revised and add more comprehensive concepts there.

Best regards

Author Response

Dear reviewer,

The document with the answers is attached.

Best regards

Reviewer 2 Report

Comments.

Row 39 – There are also animals slaughtered at 18 months (intensive system)

Row 121, 130, 138, 143 – It could be interesting to know more about the breeds reared in the 4 experimental farms (only Murrah x Mediterranean buffaloes?), and the size of the farms (hectares and number of heads)

Table 2 – The note S=M<N  in the column Local (L) it needs an explication, at bottom or in the text

Table 3 – This table seems too big. I suggest to split the table in 2 parts (1 fatty acid composition, 2 partial sum of fatty acids and fatty acid nutritional indices)

Table 3 – The notes in the columns Local (L) and Period (P) they need and explication, mainly to understand what it means PS an PC, but also S>M=N, M>S=N, etc.

Author Response

(The authors gave the same response as above.)

Reviewer 3 Report

Dear Authors

the manuscript you have presented addresses an important aspect for the evaluation of the effect of the different production systems on the qualities of water buffalo meat and of the reflecti on human health, this offers important information that can be useful for the qualitative improvement of the meat and therefore also for the consumption of red meat. The manuscript deserves to be published, however before publication it needs a revision to improve the quality of the presentation. 

Specific comments

Line 85: [22,23], spacing;

Lines 87-89: rewrite the sentence ".... carotene), also evaluating the effect of seasonal variations (dry vs rainy periods).";

Line 94: delete "x dry and rainy seasonal periods";

Line 102: specify that we are talking about the 3 extensive systems;

Lines 108-109: "in this ...... (feedlot)." it should be eliminated, it is a repetition; Lines 109-113: should be moved above line 95 after ".... section.";

Lines 137-138: "wet brewers' residual grains", enter more information;

Line 141: On line 104 you state that animals in the intensive system are slaughtered at 18 months of age, while you now state that animals are housed in barns at 18 months of age. I assume that after housing there is a finishing period, consequently the slaughter age will not be 18 months. Check and correct by providing more information;

Line 239: "Murrah × Mediterranean crossbred buffaloes", this information should be reported in paragraph 2.1;

Lines 242-247: You start by talking about the chemical composition of diets in extensive farms, then move on to the composition of fatty acids and also include the results of intensive systems. Rewrite including all significant results;

Line 244: "for example", does not seem to me the correct term. Rewrite and improve.

Table 1: The title of the table could be "Chemical composition of diets" reporting the rest of the information in the explanatory notes. Improve the structure of the table, inserting abbreviations in the first three lines, reporting the explanations between the explanatory notes. The unit of measurement for fatty acids should be% / total FA, check and correct. In Materials and Methods the determination of lipids, α-Tocopherol, γ-Tocopherol and γ – Tocotrienol is not reported, it correctly implements the Materials and Methods section or eliminates the results from the table;

Table 2: Improve the structure of the table by following the guidelines given for the table 1. Implement the notes to improve the readability of the table. Specify what you want to indicate when (S = M <N) is reported in the columns of the P-value. In the paragraph relating to statistical analysis, reference is made to production systems and seasonal variations as factors considered; what do you mean by "Local (L)"? Implement the statistical analysis section and explanatory notes of the table.

Table 3: follow the instructions given for table 2. "DMA", "Other" and "C18: 1 (1)", specify;

Lines 287-298: move before table 3 eliminating all "(Table 3)";

Lines 299-352: delete all "(Table 3)". The results are reported correctly but the exposure needs to be improved to make the reading smoother.

Lines 353-495: the discussions are well articulated but almost all the bibliographic references reported in support (from 39 to 60) relate to the effects of the composition of fatty acids on human health. It is necessary to implement the bibliography to support the results obtained in relation to the production systems, also drawing on results obtained on other species of ruminants ( https://doi.org/10.1080/19476337.2020.1842503 https://doi.org/10.1080/19476337.2020.1762746 );

Best regards

Author Response

(The authors gave the same response as above.)

Round 2

Reviewer 1 Report

The authors improved the quality of the MS and I suggest authors reading one more time to fix few language errors. Then, it would be ready for the final steps for acceptance.

  • Point 20, please summarise and shortly mention these explanations in the statistical analysis section that why you picked up this approach.
  • Line 198, please revise.
  • My comment about the common name and scientific name was to check: first time in the MS common name (Scientifc name) and rest of the MS, only common name.
  • Point 31: You repeated the result in the discussion section. I suggest you combine the result and discussion sections or revise this section.

Point 31: We have reviewed the discussion. We made some references to the results in order to be able to discuss them with more consistency.

It seems that you did not understand. I said you repeated the results in the Discussion section. Please revise the Discussion section from this point.

  • Please check errors in font and size of texts.

Author Response

Dear Reviewer,

Reviewer 3 Report

Dear Authors
thanks for considering my comments. The manuscript is now more complete and has a greater value, i report the last considerations:
- Line 277, spacing;
- Tables 1, 2, 3 and 4, the corrections made to the tables are exhaustive, but I would avoid reporting the climatic data in the tables. A specific paragraph could be inserted in the text in which to report the climatic data;
- Lines 425-430, you affirm that the lipid content of the diet is similar between the intensive system (9.77) and the extensive system (2.88-5.17), I think a bit forced as a statement. Furthermore, you state that the lipid content of the meat does not show differences between the two production systems as a consequence of the similar lipid content of the diets. In Maniaci et al. (2020) (https://doi.org/10.1080/19476337.2020.1762746) the lipid content showed differences in young bulls reared on pasture and in stable. Probably the homogeneous slaughter weight between the two production systems neutralized the different lipid levels of the diet. You could point out the differences with the mentioned work by supporting them. 

Best regards

Author Response

Dear Reviewer,
